# Unusual dominant genotype NIA1 of *Enterocytozoon bieneusi* in children in Southern Xinjiang, China

**Meng Qi[1,2], Fuchang Yu[1], Aiyun Zhao[2], Ying Zhang[2], Zilin Wei[2], Dongfang Li[1], Longxian Zhang**[1]*

**1** College of Animal Science and Veterinary Medicine, Henan Agricultural University, Zhengzhou, P. R. China, **2** College of Animal Science, Tarim University, Alar, P. R. China

* zhanglx8999@henau.edu.cn

**Data Availability Statement:** The datasets generated for this study can be found in the GenBank under the accession numbers MN136770–MN136778.

## Abstract

*Enterocytozoon bieneusi* is the mainly pathologies or intestinal disorders that causes approximately 90% of reported cases of human microsporidiosis. To understand the prevalence and genotype distribution of *E. bieneusi* in the Xinjiang Uygur Autonomous Region, China, 609 fecal samples were collected from children in kindergarten in Southern Xinjiang and screened for this pathogen by PCR and sequencing of the internal transcribed spacer (ITS). Thirty-six fecal samples (5.9%, 36/609) were positive for *E. bieneusi*, with the highest prevalence observed in children from Yopurga (17.5%, 11/63). Nine genotypes were identified, of which six were known (A, CHN6, D, EbpA, KB-1, and NIA1) and three were novel (CXJH1, CXJH2 and CXJH3). Genotype NIA1 was most prevalent (52.8%, 19/36), followed by genotypes D (16.7%, 6/36), A (8.3%, 3/36), and EbpA (8.3%, 3/36). The remaining five genotypes were detected in one sample each. Phylogenetic analysis revealed that the *E. bieneusi* isolates clustered into two groups, one consisting of six genotypes (Group 1: A, CXJH1, D, EbpA, KB-1, and NIA1) and another consisting of three genotypes (Group 2: CHN6, CXJH2, and CXJH3). Our results confirmed that infection of *E. bieneusi* unusual dominant genotype NIA1 occurs in children in Xinjiang, China. Further epidemiological studies must be conducted to clarify potential sources of *E. bieneusi* infection in this area.

## Author summary

This study reports the infection rates and genetic characteristics of *Enterocytozoon bieneusi* in 609 children in kindergarten in Southern Xinjiang, China. All samples were screened for this pathogen with PCR, based on the internal transcribed spacer (ITS) of *E. bieneusi*. Thirty-six fecal samples (5.9%, 36/609) were positive, with the highest prevalence observed in children from Yopurga (17.5%, 11/63). Three novel genotypes were identified (CXJH1, CXJH2 and CXJH3). Phylogenetic analysis revealed that the *E. bieneusi* isolates clustered into two groups: Group 1 (A, CXJH1, D, EbpA, KB-1, and NIA1) and Group 2 (CHN6, CXJH2, and CXJH3). Genotype NIA1 used to be detected in HIV-positive patients, however, it was most prevalent (52.8%, 19/36) among the nine genotypes

**Funding:** This work was supported in part by the National Natural Science Foundation of China (31860699), the National Key Research and Development Program of China (2017YFD0500405, 2017YFD0501305), and the Program for Young and Middle-Aged Leading Science, Technology, and Innovation of Xinjiang Production & Construction Group (2018CB034). The funders had no role in study design, data collection and analysis, decision to publish, or preparation of the manuscript.

**Competing interests:** The authors have declared that no competing interests exist.

identified in this study. Additionally, we confirmed the zoonotic potential of *E. bieneusi* genotype D and this is the first report of human infection by *E. bieneusi* genotypes KB-1 and CHN6.

## Introduction

Microsporidia are fungal obligate intracellular pathogens. Among at least 10 genera including 17 species infecting humans, *Enterocytozoon bieneusi* is the most frequently detected pathogen causing microsporidiosis [1]. Infection by *E. bieneusi* may have no symptoms or may cause persistent diarrhea, vomiting and a wasting syndrome, dissemination and presence in other organs, particularly in immunocompromised individuals such as human immunodeficiency virus (HIV)-infected individuals, organ transplant recipients or cancer patients. Travelers, children, and older adults are also at risk [2].

Currently, analysis of polymorphisms within the ribosomal internal transcribed spacer (ITS) is most commonly used for genotyping of *E. bieneusi*. Over 474 ITS genotypes have been identified, of which more than 96 have been detected in humans [3, 4]. Phylogenetic analysis indicated that these genotypes clustered into 11 groups (Group 1–11). Most *E. bieneusi* isolates causing human infections belonged to the zoonotic Group 1 [5]. Nevertheless, there is an increasing number of reports have been showing that some Group-2-genotypes (I, J, BEB4 and BEB6) firstly detected in livestock or wild animals can also infect humans [3, 4]. Genotypes of other groups appear to be more host-specific, suggesting a low level of potential for zoonotic or cross-species transmission. However, this hypothesis needs to be confirmed because little information is available regarding these groups [4].

In 2011, the first report emerged of *E. bieneusi* infecting humans in Jilin, China [6], with 22.5% (9/40) of children with diarrheal disease testing positive for this pathogen. Over the last decade, the prevalence and genotype distribution of *E. bieneusi* in humans in China have been reported by several studies [6–15]. The study participants had widely variable characteristics (children with or without diarrhea, HIV-positive or -negative individuals, and other immuno-compromised patients) [6–15] and the studies were conducted in diverse geographical locations (Jilin, Heilongjiang, Guangxi, Henan, Shanghai, Hubei, Chongqing, and Yunnan). However, no data are available regarding the prevalence and genetic characteristics of *E. bieneusi* isolates in the Xinjiang Uygur Autonomous Region (hereafter referred to as Xinjiang), China. Herein, we conducted a molecular epidemiological study of *E. bieneusi*, via sequencing of the ITS region, to understand the prevalence and genotypes of this pathogen in children in Xinjiang.

## Materials and Methods

### Ethics statement

All procedures complied with the ethical standards of the relevant national and institutional committees on human experimentation and with the principles laid out in the Helsinki Declaration of 1975, as revised in 2013. The study was approved by the Institutional Review Board of Henan Agricultural University (Approval No: IRC-HENAU-20160224-05). Written informed consent for participation in the study was obtained from the parent or guardian of each child after they were informed of the purposes and procedures of the study. Participants with possible presence of parasites received appropriate treatment according to local policies and national guidelines.

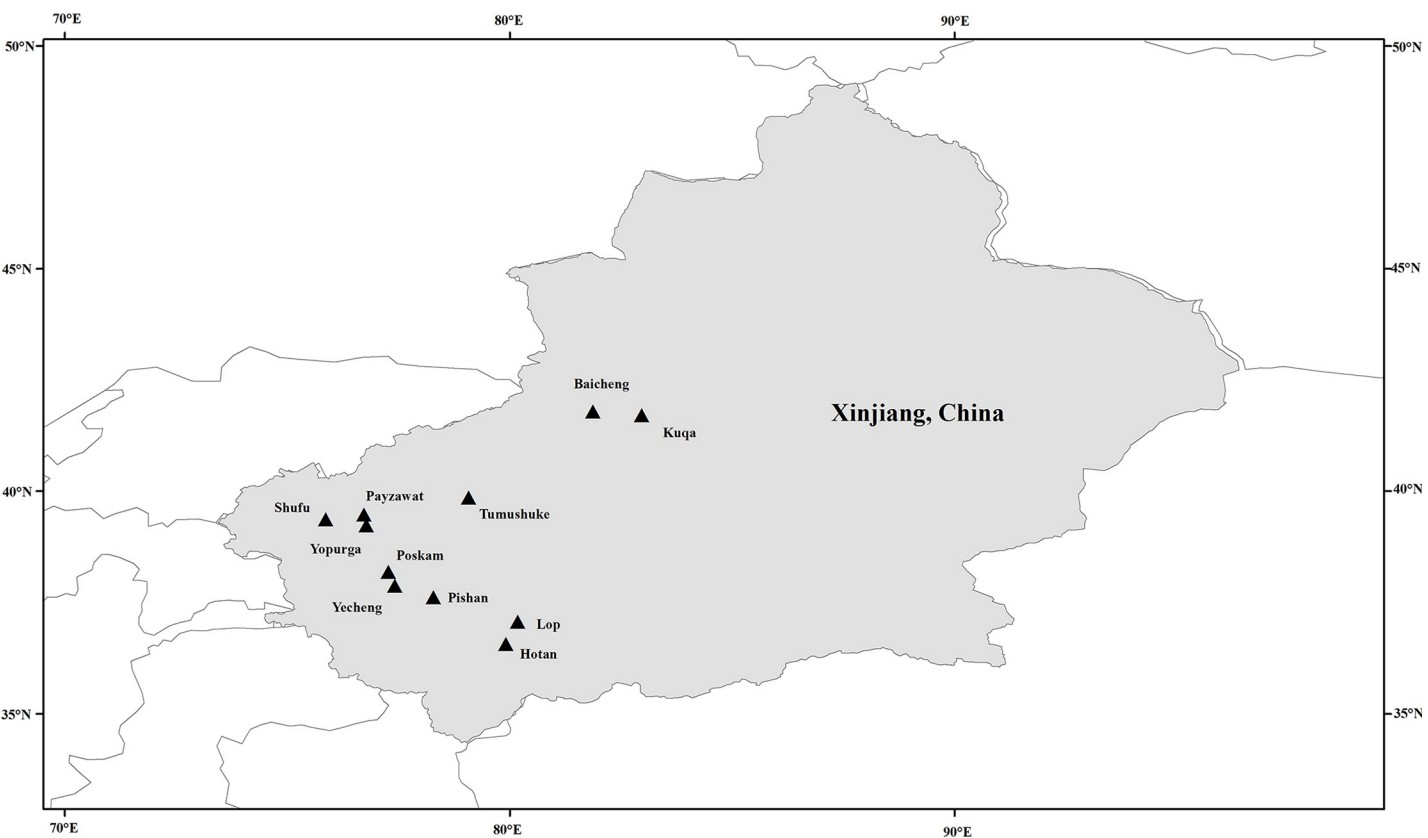

**Fig 1. Geographic map of the sampling locations in Xinjiang, China.** The figure was originally designed by the authors under the software ArcGIS 10.2. The original vector diagram imported in ArcGIS was adapted from Natural Earth (http://www.naturalearthdata.com).

## Sample collection

The study was conducted from August 2017 to January 2019. Fresh stool samples were collected from children (age range: 2 to 6 years) at kindergartens in 11 counties in Southern Xinjiang (Fig 1). After being informed of the study purpose and procedures by kindergarten staff, parents or guardians who agreed to their children's participation were given a plastic fecal collector labeled with a unique number. Fresh stool samples were collected in the morning. The fecal collector was then marked with the date and the participant's name, age, and sex. No diarrhea was observed during sampling. In total, 609 fecal samples were collected, transported to the laboratory and stored at 4˚C.

## DNA extraction and PCR amplification

Fresh fecal samples were processed within 24 h of transport to the laboratory. Genomic DNA was extracted from ~200 mg of each stool sample using the *E.Z.N.A* stool DNA kit (Omega Bio-tek, Norcross, GA) in accordance with the manufacturer's instructions. Extracted DNA was stored at -20˚C until PCR was performed.

All samples were screened for the presence of *E. bieneusi* using a nested PCR targeting a ~390-bp fragment of the ITS in accordance with the method described by Sulaiman et al. [16] with some optimization. PCR was performed in 25-μl reaction mixtures consisting of 2 μL of DNA preparation (or first PCR product), 1× PCR buffer, 200 μM dNTPs, 3 mM $MgCl_2$, 260 nM primers, and 1.5 units of *Taq* DNA polymerase (Takara, Tokyo, Japan). PCR was

performed in an Applied Biosystems 2720 thermal cycler (Applied Biosystems, Foster City, USA) using a program of 35 cycles, each consisted of denaturation at 94˚C for 35 s, annealing at 55˚C for 45 s, and extension for 45 s at 72˚C; an initial denaturation step of 95˚C for 5 min and a final extension step of 72˚C for 10 min were also included. Outer primers were EBITS3 (5'-GGTCATAGGGATGAAGAG-3') and EBITS4 (5'-TTCGAGTTCTTTCGCGCTC-3'); EBITS1 (5'-GCTCTGAATATCTATGGCT-3') and EBITS2.4 (5'-ATCGCCGACGGATC CAAGTG-3') were used as nested primers for secondary PCR. Reagent-grade water and *E. bieneusi*-positive DNA (dairy cattle-derived genotype I DNA) were used as negative and positive controls, respectively.

## Sequence and phylogenetic analyses

Positive secondary PCR products were sequenced bidirectionally by GENEWIZ (Suzhou, China). The obtained sequences were assembled and edited using DNASTAR Lasergene Edit-Seq version 7.1.0 (http://www.dnastar.com/) and aligned with reference sequences downloaded from GenBank in the software Clustal X version 2.1 (http://www.clustal.org/).

The typical length of *E. bieneusi* ITS nucleotide sequence was 243 bp. Phylogenetic relationships were inferred by constructing a phylogenetic tree using Bayesian inference (BI) and the Monte Carlo Markov Chain method in MrBayes v3.2.6 (http://mrbayes.sourceforge.net/) and Fig-Tree v1.4.4 (http://tree.bio.ed.ac.uk/software/figtree/). Posterior probabilities were estimated based on 1,000,000 generations with four simultaneous tree building chains, with trees saved every 100[th] generation. A 50% majority rule consensus tree for each analysis was constructed based on the final 75% of trees generated using BI.

## Statistical analyses

Prevalence of *E. bieneusi* with an associated 95% confidence interval (CI) was calculated using IBM SPSS Statistics (www.ibm.com/products/spss-statistics). Fisher's exact test was used to compare the prevalence of *E. bieneusi* in different groups. A two-sided *p*-value of ≤0.05 was considered statistically significant.

## Nucleotide sequence accession numbers

All nucleotide sequences obtained in this study were deposited in GenBank under accession numbers MN136770 to MN136778.

# Results

## Prevalence of *E. bieneusi*

Of the 609 fecal samples analyzed by nested PCR, 36 were positive for *E. bieneusi*, yielding an overall prevalence 5.9% (95% CI 4.0–7.9%). *E. bieneusi*-positive samples were detected in eight of the 11 counties involved in this study (Table 1). The highest prevalence was observed in Yopurga (17.5%, 11/63; 95% CI: 7.3–27.7%) followed by Baicheng (13.0%, 3/23; 95% CI: 0–29.0%). Prevalence ranged from 1.6% to 9.0% in the remaining six counties, and *E. bieneusi* was not detected in Hotan, Poskam or Kuqa (Table 1). Differences in *E. bieneusi* prevalence among counties in which the parasite was detected were statistically significant (*p* = 0.043).

*E. bieneusi* was detected in 6.7% (20/299, 95% CI: 3.7%–9.7%) and 5.2% (16/310, 95% CI: 2.5%–7.8%) of samples from boys and girls, respectively. Prevalence differences between gender groups were not significant (*p* > 0.05) (Table 2).

**Table 1. Prevalence and genotype distribution of *E. bieneusi* among children in different counties of southern Xinjiang.**

| County | No. of positives/ No. of samples | Prevalence (%) (95% CI) | Genotype (n) |
|---|---|---|---|
| Tumushuke | 1/62 | 1.6 (0–5.6) | NIA1 (1) |
| Payzawat | 2/25 | 8.0 (0–20.6) | NIA1 (2) |
| Shufu | 3/48 | 6.3 (0–14.1) | D (1), NIA1 (2) |
| Yopurga | 11/63 | 17.5 (7.3–27.7) | CXJH1 (1), D (4), NIA1 (6) |
| Yecheng | 8/89 | 9.0 (2.5–15.5) | A (3), D (1), NIA1 (4) |
| Hotan | 0/80 | 0 | |
| Baicheng | 3/23 | 13.0 (0–29.0) | CHN6 (1), CXJH2 (1), CXJH3 (1) |
| Poskam | 0/35 | 0 | |
| Kuqa | 0/38 | 0 | |
| Pishan | 3/37 | 8.1 (0–18.3) | EbpA (2), NIA1 (1) |
| Lop | 5/109 | 4.6 (0.2–9.0) | EbpA (1), KB-1 (1), NIA1 (3) |
| Total | 36/609 | 5.9 (4.0–7.9) | A (3), CHN6 (1), CXJH1 (1), CXJH 2 (1), CXJH 3 (1), D (6), EbpA (3), KB-1 (1), NIA1 (19), |

## Genotype distribution of *E. bieneusi*

Nucleotide sequences of the ITS were analyzed for all 36 *E. bieneusi* PCR-positive samples. The results revealed nine distinct genotypes, including six known genotypes (A, CHN6, D, EbpA, KB-1, and NIA1) and three novel genotypes (CXJH1, CXJH2 and CXJH3) (Table 1). NIA1 was the most frequently-detected genotype (n = 19), followed by genotypes D (n = 6), A (n = 3), and EbpA (n = 3). The remaining genotypes were detected in one sample each.

## Phylogenetic analysis

Compared with genotype NIA1 (EF428628), the novel genotype CXJH1 displayed a single nucleotide polymorphism (SNP) at position 17 (G→A) of the ITS sequence. Genotypes CXJH2 and CXJH3 were characterized by two SNPs each (A43G and T201C in CXJH2; A3G and A43G for CXJH3) compared with genotype CHN6 (HM992514). Phylogenetic analysis was performed using the ITS region sequences to understand the genetic relationships among the novel *E. bieneusi* genotypes identified in this study and known genotypes. Genotypes NIA1, D, KB-1, A, EbpA, and the novel genotype CXJH1 clustered into one group (Group 1) and the remaining three genotypes (CHN6, CXJH2, and CXJH3) clustered in Group 2 together with the zoonotic genotypes I, J, and BEB6 (Fig 2).

## Discussion

The average prevalence of *E. bieneusi* measured in humans in China is approximately 5.8% [17], with the highest prevalence (22.5%, 9/40) reported in Changchun [6] and the lowest (0.2%, 1/500) in Wuhan [12]. The prevalence of *E. bieneusi* detected in the present study was 5.9%, and similar results have been reported in several previous studies. For example, in a previous study, the prevalence of *E. bieneusi* in children of various age categories and clinical presentations was 7.5% (19/225) [10]. Similar prevalence was also reported in a contrast study

**Table 2. Prevalence and genotype distribution of *E. bieneusi* among children in southern Xinjiang by gender.**

| Gender | No. of positives/ No. of samples | Prevalence (%) (95% CI) | Genotype (n) |
|---|---|---|---|
| Boy | 20/299 | 6.7 (3.7–9.7) | A (2), CHN6 (1), D (3), EbpA (1), NIA1 (13) |
| Girl | 16/310 | 5.2 (2.5–7.8) | A (1), CXJH1 (1), CXJH 2 (1), CXJH 3 (1), D (3), EbpA (2), KB-1 (1), NIA1 (6), |

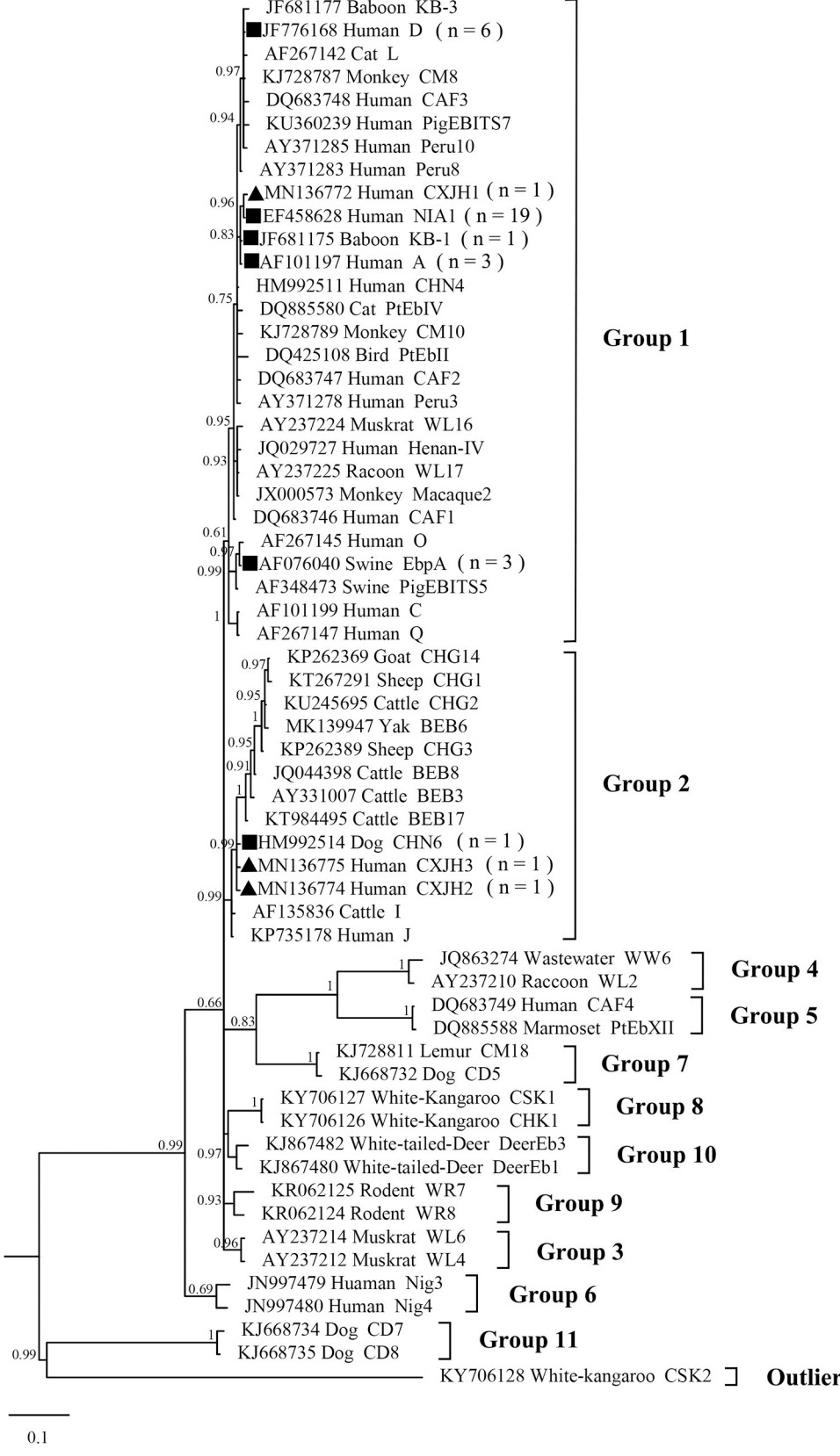

**Fig 2. Bayesian phylogenetic analysis of *E. bieneusi* ITS sequences.** Statistically significant posterior probabilities are indicated at branches. Sample names include GenBank accession number followed by host and then genotype designation. The *E. bieneusi* genotype CSK2 (KY706128) from red kangaroo was used as the outgroup. Known and novel genotypes identified in this study are indicated by squares and triangles, respectively.

conducted in Henan Province in China, in which 683 HIV-negative and 683 HIV-positive individuals were screened for the presence of *E. bieneusi*; 4.2% (29/683) and 5.7% (39/683), respectively, tested positive for this parasite [8]. However, higher prevalence has also been reported in China, such as in a study which found that 11 of 93 (11.8%) fecal specimens collected from children with diarrheal disease in Chongqing were positive for *E. bieneusi* [14]. Large differences in *E. bieneusi* prevalence in different regions in China have been reported by many studies, but the reasons underlying these differences remain unclear. *E. bieneusi* prevalence is associated with many factors, including but not limited to the immune status of the host, healthcare level, economic status and living conditions [13]. To determine the specific factors responsible for prevalence differences, more epidemiological investigations will need to be conducted.

In this study, *E. bieneusi* NIA1 was the most frequently-detected genotype (52.8%, 19/36). NIA1 was first detected in an HIV-positive patient in Niamey, Niger in 2007 [18]. Subsequently, NIA1 was detected in two hospitalized Acquired Immune Deficiency Syndrome (AIDS) patients in the Democratic Republic of the Congo in 2010 [19], one HIV-positive patient in China in 2011 [20], and one HIV-positive patient in the Democratic Republic of the Congo in 2012 [21]. Surprisingly, *E. bieneusi* NIA1 has only been identified in HIV-positive patients in previous studies, but was the dominant genotype (52.8%, 19/36) in children in Southern Xinjiang in this study. Whether there is a correlation between the genotype NIA1 and the immune status of local children, or a geographical role in the distribution of this genotype, requires further investigation.

Genotype D was the second most prevalent genotype detected in this study. According to the few reports of human infection by this genotype in China, genotype D was frequently detected in children with diarrhea, HIV-positive patients, and HIV-negative individuals in Shanghai, Henan, Hubei, and Guangxi [7, 8, 11, 12]. Additionally, genotype D has also been detected in humans in at least 21 countries worldwide [4] and in a vast range of animal species including companion animals, livestock, wildlife, rodents, and birds [4, 13]. In addition, genotype D has been detected in river water, wastewater, and combined sewer overflow in China [22–24] as well as in wastewater in Spain and Tunisia [25, 26]. Therefore, the zoonotic potential of *E. bieneusi* genotype D is established, and its transmission via the indirect fecal–oral route is a strong possibility.

Among the remaining known genotypes, KB-1, CHN6, and EbpA were first identified in nonhuman primates (NHPs), dogs, and pigs, respectively [6, 27, 28]. Genotype EbpA has been detected in many hosts, including NHPs, sheep, goats, deer, dairy cattle, and humans [4]. Moreover, in one of our previous studies genotypes D and EbpA have been detected in pigs in the same geographical area of Xinjiang (China) [29]. This may indicate the animal origin of *E. bieneusi* in this area. Since all the samples involved here were collected from counties in rural area, the role of livestock in the circulation of this pathogen is nonnegligible in Xinjiang, China. Interestingly, ours is the first report of human infection by *E. bieneusi* genotypes KB-1 and CHN6.

According to the phylogenetic tree inferred in our study, the novel genotype CXJH1 clustered together with genotypes KB-1, A, and NIA1 in Group 1, potentially reflecting the zoonotic potential of these four genotypes. In contrast, genotypes CXJH2 and CXJH3 clustered together with genotype CHN6 in Group 2. Previous studies suggested that Group 2 was

composed of *E. bieneusi* genotypes identified in ruminants. However, Group 2 genotypes have also increasingly been detected in human samples, suggesting their zoonotic potential [7].

In conclusion, the results of our study showed that *E. bieneusi* infection occurs in children in Xinjiang, China. Unusually, genotype NIA1 was identified as the most prevalent genotype among the fecal samples examined. Further epidemiological studies need to be conducted to confirm the animal origin of *E. bieneusi* infection in this area.

## Supporting information

**S1 Checklist. STROBE Statement.** Checklist of items that should be included in reports of observational studies.
(DOCX)

## Acknowledgments

We thank Liwen Bianji, Edanz Editing China (www.liwenbianji.cn/ac), for editing the English text of a draft of this manuscript.

## Author Contributions

**Conceptualization:** Meng Qi, Fuchang Yu, Longxian Zhang.

**Formal analysis:** Meng Qi, Fuchang Yu, Dongfang Li, Longxian Zhang.

**Funding acquisition:** Longxian Zhang.

**Investigation:** Aiyun Zhao, Ying Zhang, Zilin Wei.

**Methodology:** Meng Qi, Fuchang Yu, Longxian Zhang.

**Project administration:** Meng Qi, Longxian Zhang.

**Resources:** Aiyun Zhao, Ying Zhang, Zilin Wei.

**Supervision:** Meng Qi, Longxian Zhang.

**Validation:** Meng Qi, Fuchang Yu, Aiyun Zhao.

**Visualization:** Fuchang Yu, Dongfang Li.

**Writing – original draft:** Meng Qi, Fuchang Yu.

**Writing – review & editing:** Meng Qi, Fuchang Yu, Longxian Zhang.

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
