## [Decision Letter · Decision Letter 0]

31 Jan 2020

Dear Dr. Zhang,

Thank you very much for submitting your manuscript "Unusual dominant genotype NIA1 of Enterocytozoon bieneusi in children in Southern Xinjiang, China" for consideration at PLOS Neglected Tropical Diseases. As with all papers reviewed by the journal, your manuscript was reviewed by members of the editorial board and by several independent reviewers. In light of the reviews (below this email), we would like to invite the resubmission of a significantly-revised version that takes into account the reviewers' comments. 

We cannot make any decision about publication until we have seen the revised manuscript and your response to the reviewers' comments. Your revised manuscript is also likely to be sent to reviewers for further evaluation.

Sincerely,

Abdallah M. Samy, PhD

Guest Editor

Todd Reynolds

Deputy Editor

Editor comments to authors: I invited four reviews for your manuscript; however, we received only three reviews and experienced a delay in receiving the fourth review. My decision is to terminate the review task of this fourth review to avoid any further delay on this manuscript. Please consider all comments addressed by reviewers below before submitting a revised version of your manuscript. Thanks! AMS

Reviewer's Responses to Questions

**Key Review Criteria Required for Acceptance?**

**Methods**

-Are the objectives of the study clearly articulated with a clear testable hypothesis stated?

-Is the study design appropriate to address the stated objectives?

-Is the population clearly described and appropriate for the hypothesis being tested?

-Is the sample size sufficient to ensure adequate power to address the hypothesis being tested?

-Were correct statistical analysis used to support conclusions?

-Are there concerns about ethical or regulatory requirements being met?

Reviewer #1: -Are the objectives of the study clearly articulated with a clear testable hypothesis stated? YES

-Is the study design appropriate to address the stated objectives? YES

-Is the population clearly described and appropriate for the hypothesis being tested? NO

-Is the sample size sufficient to ensure adequate power to address the hypothesis being tested? YES

-Were correct statistical analysis used to support conclusions? YES

-Are there concerns about ethical or regulatory requirements being met? NO

"Comments to the Author":

Keywords: 

- Add: novel genotypes. 

Abstract: 

- Lines 16-17: “Enterocytozoon bieneusi causes approximately 90% of reported cases of human microsporidiosis”. Add: “mainly pathologies or intestinal disorders”. 

Author summary: 

- Lines 40-41: “Additionally, we demonstrated the zoonotic potential of E. bieneusi genotype D..”. Change: I prefer to use “we confirmed” or “we contrasted”. 

- Comment: The authors must mention the groups 1 and 2 for grouping the nine identified genotypes and highlight the three novel genotypes. 

Introduction: 

- Line 48: “may cause persistent diarrhea, vomiting and a wasting síndrome”. Add: “dissemination and presence in other organs”. 

- Lines 57-58. Comment: The zoonotic role of group 2 and mainly of group 1 should be highlighted. Finally, cite risk factors, sources of infection and animal hosts (reservoirs). 

- Line 64: “in China have been reported by several studies”. Add: references [6-14]?. 

Materials and Methods:

- Ethics statements: Lines 82-83. “Participants infected with parasites received 83 appropriate treatment according to local policies and national guidelines”. Comment: It is unknown if patients have an active infection or are spores passing through the intestine without causing a real infection in the patient. I prefer to remove “Participants infected” and put “Participants with possible presence of parasites” or “Positive participants in the study with parasites”. 

- Sample collection: Lines 86-87. In my opinion it would be very important to indicate if the counties studied are in urban or rural areas. This information would help a greater knowledge of the circulation of different genotypes according to the type of area and the presence or absence of animal hosts as a source of infection. 

- Sample collection: Line 91. “No diarrhea was observed during sampling”. Comment: It should be indicated if children have any other signs or symptoms, for example, weight loss, malnutrition, fever, etc. or even some other associated pathology (if this would be possible). 

- DNA extraction and PCR amplification. The authors should provide some additional nested PCR information. For example, the pairs of primers used, temperature cycles, independently of the bibliographic references.

Reviewer #2: 1.the sample collection is from 11 counties of Xinjiang ,China, suggest to use a map to fix location.

2."the studies were conducted in diverse geographical locations (Jilin, Heilongjiang, Guangxi, Henan, Shanghai, Hubei, and Chongqing)", need to supply more references, such as PLoS Negl Trop Dis 13(5): e0007356. https://doi.org/10.1371/journal.pntd.0007356.

Reviewer #3: (No Response)

**Results**

-Does the analysis presented match the analysis plan?

-Are the results clearly and completely presented?

-Are the figures (Tables, Images) of sufficient quality for clarity?

Reviewer #1: -Does the analysis presented match the analysis plan?. 

Yes.

-Are the results clearly and completely presented?

Phylogenetic analysis: Lines 163-164. “(CHN6, CXJH2, and CXJH3) clustered together with the zoonotic genotypes I, J, and BEB6 (Figure 1)”. Add: group 2.

-Are the figures (Tables, Images) of sufficient quality for clarity?

Table 1: Indicate the urban or rural character of the counties studied.

Reviewer #2: 1.the line 141, I think that it make a mistake, "age groupe"? If so, please add it.

Reviewer #3: (No Response)

**Conclusions**

-Are the conclusions supported by the data presented?

-Are the limitations of analysis clearly described?

-Do the authors discuss how these data can be helpful to advance our understanding of the topic under study?

-Is public health relevance addressed?

Reviewer #1: -Are the conclusions supported by the data presented? Review (comments below)

-Are the limitations of analysis clearly described? YES

-Do the authors discuss how these data can be helpful to advance our understanding of the topic under study? NO

-Is public health relevance addressed? YES

The authors should modify the conclusions based on the new information provided in the Discussion.

For example:

- Line 201. Remember: In my opinion it would be very important to indicate if the counties studied are in urban or rural areas. This information would help a greater knowledge of the circulation of different genotypes according to the type of area and the presence or absence of animal hosts as a source of infection. 

- The authors have recently published the following article that should be discussed in this section and included in the bibliography. “Li D-F, Zhang Y, Jiang Y-X, Xing J-M, Tao D-Y, Zhao AY, Cui Z-H, Jing B, Qi M and Zhang L-X (2019). Genotyping and Zoonotic Potential of Enterocytozoon bieneusi in Pigs in Xinjiang, China. Front. Microbiol. 10:2401. doi: 10.3389/fmicb.2019.02401”. 

It is very important because the second genotype (D) and the fourth (EbpA) identified in children are those detected in pigs in the same geographical area of Xinjiang (China). 

Therefore, it could reinforce the role or zoonotic character of some of the genotypes found in children and the importance of the geographical area (urban or rural).

Reviewer #2: this study conclusions are proper. the data of analysis is correct. the discuss can help to understanding the topics.

Reviewer #3: (No Response)

**Editorial and Data Presentation Modifications?**

Reviewer #1: (No Response)

Reviewer #2: the objectives of study are very clear, and the sdudy design is appropriate toaddress the objectives.

Reviewer #3: (No Response)

**Summary and General Comments**

Reviewer #1: Discussion: 

- Line 184: “as in a study which found that 11 of 93 fecal specimens collected from children with”. Add: percentage (%). 

- Line 199: “dominant genotype in children in Southern Xinjiang in this study”. Add: percentage (%). 

- Line 201. Remember: In my opinion it would be very important to indicate if the counties studied are in urban or rural areas. This information would help a greater knowledge of the circulation of different genotypes according to the type of area and the presence or absence of animal hosts as a source of infection. 

- Line 202: “Genotype D was the second most prevalent genotype detected in this study”. Add: percentage (%). 

- The authors have recently published the following article that should be discussed in this section and included in the bibliography. “Li D-F, Zhang Y, Jiang Y-X, Xing J-M, Tao D-Y, Zhao AY, Cui Z-H, Jing B, Qi M and Zhang L-X (2019). Genotyping and Zoonotic Potential of Enterocytozoon bieneusi in Pigs in Xinjiang, China. Front. Microbiol. 10:2401. doi: 10.3389/fmicb.2019.02401”. 

It is very important because the second genotype (D) and the fourth (EbpA) identified in children are those detected in pigs in the same geographical area of Xinjiang (China). 

Therefore, it could reinforce the role or zoonotic character of some of the genotypes found in children and the importance of the geographical area (urban or rural). 

References: 

- Add: Li D-F, Zhang Y, Jiang Y-X, Xing J-M, Tao D-Y, Zhao A-Y, Cui Z-H, Jing B, Qi M and Zhang LX (2019). Genotyping and Zoonotic Potential of Enterocytozoon bieneusi in Pigs in Xinjiang, China. Front. Microbiol. 10:2401. doi: 10.3389/fmicb.2019.02401.

Reviewer #2: This study reports the zoonotic potential of E. bieneusi genotype D . this study is also the first report of human infection by E. bieneusi genotypes KB-1 and CHN6 in China. This is an important significance .

Reviewer #3: (No Response)

PLOS authors have the option to publish the peer review history of their article (what does this mean?). If published, this will include your full peer review and any attached files.

Reviewer #1: Yes: Dr. Fernando Izquierdo Arias. 

Área de Parasitología e Inmunología.

Facultad de Farmacia.

Universidad CEU-San Pablo.

Urbanización Montepríncipe.

28668 Boadilla del Monte, Madrid (Spain)

Teléfono: +34-91-372-47-00 (ext. 15275)

Fax: 91-351-04-96

e-mail: ferizqui@ceu.es

https://orcid.org/0000-0003-0367-007X

Publons: AAE-3073-2019

Reviewer #2: No

Reviewer #3: No
---

## [Decision Letter · Decision Letter 1]

10 Apr 2020

Dear Dr. Zhang,

Thank you very much for submitting your manuscript "Unusual dominant genotype NIA1 of Enterocytozoon bieneusi in children in Southern Xinjiang, China" for consideration at PLOS Neglected Tropical Diseases. As with all papers reviewed by the journal, your manuscript was reviewed by members of the editorial board and by several independent reviewers. The reviewers appreciated the attention to an important topic. Based on the reviews, we are likely to accept this manuscript for publication, providing that you modify the manuscript according to the review recommendations. 

Please prepare and submit your revised manuscript within 7 days. If you anticipate any delay, please let us know the expected resubmission date by replying to this email.  

Sincerely,

Abdallah M. Samy, PhD

Deputy Editor

Todd Reynolds

Deputy Editor

Reviewer's Responses to Questions

**Key Review Criteria Required for Acceptance?**

**Methods**

-Are the objectives of the study clearly articulated with a clear testable hypothesis stated?

-Is the study design appropriate to address the stated objectives?

-Is the population clearly described and appropriate for the hypothesis being tested?

-Is the sample size sufficient to ensure adequate power to address the hypothesis being tested?

-Were correct statistical analysis used to support conclusions?

-Are there concerns about ethical or regulatory requirements being met?

Reviewer #1: -Are the objectives of the study clearly articulated with a clear testable hypothesis stated? YES

-Is the study design appropriate to address the stated objectives? YES

-Is the population clearly described and appropriate for the hypothesis being tested? YES

-Is the sample size sufficient to ensure adequate power to address the hypothesis being tested? YES

-Were correct statistical analysis used to support conclusions? YES

-Are there concerns about ethical or regulatory requirements being met? NO

Reviewer #3: Yes

**Results**

-Does the analysis presented match the analysis plan?

-Are the results clearly and completely presented?

-Are the figures (Tables, Images) of sufficient quality for clarity?

Reviewer #1: -Does the analysis presented match the analysis plan? YES

-Are the results clearly and completely presented? YES

-Are the figures (Tables, Images) of sufficient quality for clarity? YES

Reviewer #3: Yes

**Conclusions**

-Are the conclusions supported by the data presented?

-Are the limitations of analysis clearly described?

-Do the authors discuss how these data can be helpful to advance our understanding of the topic under study?

-Is public health relevance addressed?

Reviewer #1: -Are the conclusions supported by the data presented? YES

-Are the limitations of analysis clearly described? YES

-Do the authors discuss how these data can be helpful to advance our understanding of the topic under study? YES

-Is public health relevance addressed? YES

Reviewer #3: Yes

**Editorial and Data Presentation Modifications?**

Reviewer #1: Comment:

Question 2: - Lines 16-17: “Enterocytozoon bieneusi causes approximately 90% of reported cases of human microsporidiosis”. Add: “mainly pathologies or intestinal disorders”. Response: Thank you very much. We have revised this. Seen in Line 16. 

The authors must put all words “mainly pathologies or intestinal disorders”. In the manuscript is only "Enterocytozoon bieneusi is the mainly pathologies".

Reviewer #3: Accept

**Summary and General Comments**

Reviewer #1: Dear Authors:

I want to thank the authors for the changes done and congratulate them for the work and research presented in the article. And I encourage them to continue in the same line for clearing doubts about the circulation and prevalence of this parasite.

Reviewer #3: The author has already addressed the relevant questions and the authors have made the changes. I have just a few additional comments.

1. “unusual dominant genotype NIA1” is missing in the abstract, please add it.

2. Line 59, ‘of which 96 have been…’ should be revised to ‘of which more than 96 have been…’

3. Line 213, healthcare level also should be included here.

4. Line 251, ‘…isolates identified in ruminants.’. Revised word ‘isolates’ to ‘genotypes’.

PLOS authors have the option to publish the peer review history of their article (what does this mean?). If published, this will include your full peer review and any attached files.

Reviewer #1: Yes: Dr. Fernando Izquierdo Arias.

Reviewer #3: Yes: Jianping Cao
---

## [Editor Report · Decision Letter 2]

14 Apr 2020

Dear Dr. Zhang,

We are pleased to inform you that your manuscript 'Unusual dominant genotype NIA1 of Enterocytozoon bieneusi in children in Southern Xinjiang, China' has been provisionally accepted for publication in PLOS Neglected Tropical Diseases.

Best regards,

Abdallah M. Samy, PhD

Deputy Editor

Todd Reynolds

Deputy Editor

---

## [Editor Report · Acceptance letter]

12 Jun 2020

Dear Dr. Zhang,

We are delighted to inform you that your manuscript, "Unusual dominant genotype NIA1 of Enterocytozoon bieneusi in children in Southern Xinjiang, China," has been formally accepted for publication in PLOS Neglected Tropical Diseases.

Best regards,

Shaden Kamhawi

co-Editor-in-Chief

Paul Brindley

co-Editor-in-Chief
